

# Simultaneous measurement of NO and NO₂ by dual-channel cavity ring down spectroscopy technique

Renzhi Hu (1)†*, Zhiyan Li (1, 2) †, Pinhua Xie (1, 3, 5) *, Hao Chen (1), Xiaoyan Liu(4),
Shuaixi Liang (1, 2), Dan Wang (6), Fengyang Wang(1), Yihui Wang(1,3), Chuan Lin (1), Jianguo
Liu(1, 3, 5), Wenqing Liu(1, 3, 5)

*(1) Key Lab. of Environmental Optics and Technology , Anhui Institute of Optics and Fine Mechanics, Chinese Academy of Sciences, Hefei 230031,China.*

*(2) Science Island Branch of Graduate School, University of Science and Technology of China. Hefei 230026, China.*

*(3) School of Environmental Science and Optoelectronic Technology, University of Science and Technology of China, Hefei 230027, China*

*(4) College of Pharmacy, Anhui Medical University, 81 Meishan Road, Hefei 230032, China*

*(5) CAS Center for Excellence in Regional Atmospheric Environment, Institute of Urban Environment, Chinese Academy of Sciences, Xiamen, 361000, Fujian, China*

*(6) School of Mathematics and Physics, Anhui University of Technology, Ma an Shan 243032, China*

†These authors contributed equally to this work.
*e-mail: rzhu@aiofm.ac.cn, phxie@aiofm.ac.cn
Key words: NO$_x$; CRDS; CEAS; CL

## Abstract

Nitric oxide (NO) and nitrogen dioxide (NO₂) are relevant to air quality due to their role in tropospheric ozone (O₃) production. In China, NOx emissions are high and exhausted from on-road vehicles make up 20% of total NOx emissions. Too much NO$_x$ are harmful to the human body and animals. In order to detect the NO and NO₂ emissions on road, a dual-channel CRDS system for NO₂ and NO detection is reported. In this system, NO is converted to NO₂ by its reaction with excess O₃ in NO$_x$ channel, such that NO can be determined through the difference between two channels. The detection limits of the developed CRDS system for NO₂ and NO$_x$ measurements are estimated to be about 0.030 ppb (1σ, 1 s) and 0.040 ppb (1σ, 1 s), respectively. Considering the error sources of NO₂ absorption cross section and R$_L$ determination, the total uncertainty of NO₂ measurements is about 5%. The CRDS method is capable of measuring species with high sensitivity and accuracy. The performance of the system was validated against a chemiluminescence (CL) analyzer (42i, Thermo Scientific, Inc.) when measuring the NO₂ standard mixtures. The results of NO₂ with standard mixtures sampled showed a linear correction factor (R$^2$) of 0.99 in a slope of 1.031 ± 0.006, with an offset of (-0.940 ± 0.323) ppb. An intercomparison between the system and a cavity-enhanced absorption spectroscopy (CEAS) instrument for NO₂ measurement was also conducted alone in ambient environment. Least-squares analysis showed that the slope and intercept of the regression line are 1.042 ± 0.002 and (−0.393 ± 0.040) ppb, respectively, with a linear correlation factor of $R^2$ = 0.99. Another intercomparison conducted between the system and the CL analyzer for NO detection also showed a good agreement within their uncertainties, with an absolute shift of (0.352 ± 0.013) ppb, a slope of 0.957 ± 0.007 and a correlation coefficient of $R^2$ = 0.99. The measurements of on-road vehicle emission plumes by this mobile CRDS instrument show the different emission characteristics in the urban and



suburban areas of Hefei. The instrument provides a new method for retrieving fast variations of NO and $NO_2$ plumes.

## 1.  Introduction

In recent years, with the improvement of people's living standard, people pay more and more attention to the improvement of the living environment. Among which, the management of environmental pollution has gradually become one of the focus issues. The detection of pollutants is an important premise for environmental governance. $NO_x$ ($NO_x$= $NO$+$NO_2$) are byproducts of organic decay, natural forest fires as well as anthropogenic emission both from stationary sources (electric

power generation using fossil fuels) (Jaramillo and Muller, 2016) and mobile sources (motor vehicles and catalytic converters of most cars) (Carslaw, 2005). $NO_x$ are both primary pollutants and secondary pollutants (Crutzen, 1979), which can determine the tropospheric $O_3$ levels and lead to the formation of photochemical "smog" and the visibility decline due to the secondary aerosol formation. Furthermore, $NO_x$ are also the precursors of nitric acid (Brown et al., 2004). Moreover, $NO_x$ are harmful to the

human body and animals. Too much high $NO_x$ can damage the respiratory system and lead to pulmonary edema (Yang and Omaye, 2009). In addition, accurate $NO_2$ measurement plays a key role in accurate measurement of other species, such as organic nitrate (Thieser et al., 2016; Paul et al., 2009; Day et al., 2002) and $RO_2$ radicals (Chen et al., 2016).

During the last few years, many direct and indirect techniques for monitoring $NO_2$ have been

established. $NO_2$ concentration can be measured with chemiluminescence (CL) detection (Yuba et al., 2010; Sadanaga et al., 2008; Fahey et al., 1985), differential optical absorption spectroscopy (DOAS) (Platt et al., 1984;R. McLaren, 2010), tunable diode laser absorption spectroscopy (TDLAS) (Li et al., 2004), cavity ring-down spectroscopy (CRDS) (Castellanos et al., 2009; Fuchs et al., 2009; Osthoff et al., 2006; Fuchs et al., 2010; Brent et al., 2013 ; Hu et al.,2015), cavity enhanced absorption

spectroscopy (CEAS) (Wu et al., 2009; Gherman et al., 2008; Kasyutich et al., 2006; Wada and Orr-Ewing, 2005), cavity attenuated phase shift spectroscopy (CAPS) (Kebabian et al., 2008), laser-induced fluorescence (LIF) (Taketani et al., 2007; Matsumi et al., 2010; Sadanaga et al., 2014; Matsumoto et al., 2001) measurement, long path absorption photometer` (LOPAP) (Villena et al., 2011) and gas based sensors (Novikov et al., 2016), with CL being the most widely used for ambient in situ

sampling. CL can achieve direct measurement of NO and indirect measurement of $NO_2$. The method is based on the reaction between NO and $O_3$, which can form an electronically excited molecule of $NO_2^*$. When $NO_2^*$ reaches the ground state, it emits fluorescence which is proportional to the NO value. $NO_2$ is measured by its conversion to NO and usually heated (300 ℃ to 350 ℃) molybdenum (Mo) surfaces (Ridley and Howlett, 1974) or photolytic $NO_2$ converters like Xenon lamps or UV emitting diodes at

specific wavelength (320 nm-400 nm) are used. The CL instruments have typical $NO_2$ detection limits of 50 ppt / 1 min  (1σ)(Wang et al., 2001). CRDS,CEAS, CAPS and TDLAS relying on scanning a light source through a range of frequencies of interest are all direct absorption techniques. These techniques can achieve a high sensitivity of several seconds and a low detection limit of ppt level (Li et al., 2004; Wild et al., 2014; Gherman et al., 2008; Kebabian et al., 2008). Among these techniques,

CRDS has become a promising technique for ambient $NO_2$ detection due to its advantages of high time resolution, low detection limit as well as portability, in which pulsed (Fuchs et al., 2009) and continuous-wave (cw) (Wada and Orr-Ewing, 2005) lasers were utilized. Wada et al. (Wada and Orr-Ewing, 2005) demonstrated a cw diode CRDS system operating at 410 nm for the retrieval of $NO_2$ mixing ratios in ambient air with a detection limit of 0.1 ppb in 50 s at atmospheric pressure. Osthoff et



al. (Osthoff et al., 2006) constructed a pulsed cavity ring-down spectrometer which used a pulsed (20–100 Hz, up to 25 mJ) frequency-doubled Nd:YAG laser for the simultaneous measurements of $NO_2$, nitrate radical ($NO_3$), and dinitrogen pentoxide ($N_2O_5$) in the atmosphere. The $NO_2$ detection limit ($1\sigma$) for 1 s data was 40 ppt with an uncertainty within $\pm 4\%$ under laboratory conditions. Fuchs (Fuchs et al., 2009) used a simple, lightweight, low power, commercially available Fabry-Perot (FP) diode laser

with a center wavelength of 403.96 nm as a light source to detect NO and $NO_2$ in two separate channels. The limit of detection is 22 ppt ($2\sigma$ precision) for $NO_2$ at 1 s time resolution. Karpf (Karpf et al., 2016) used a high-power, multimode Fabry Perot (FP) diode laser with a broad wavelength range ($\Delta\lambda_{laser} \sim$ 0.6 nm) to excite a large number of cavity modes, thereby reducing the susceptibility of the detector to vibration and making it well suited for field deployment. A sensitivity of 38 ppt was achieved using an

integration time of 128 ms for single-shot detection. A number of intercomparison studies demonstrating the accuracy of these research grade instruments have been carried out (Xu et al., 2013; Dunlea et al., 2007; Villena et al., 2012) to evaluate the uncertainty of each instrument. The comparison results show that the method based on Mo converters is affected by significant interferences such as $N_2O_5$, HONO, $HNO_3$, PAN, etc. Whereas the method based on optical absorption is relatively immune

to interferences. Therefore, direct techniques are considered to be more reliable methods than the CL method for the measurement $NO_2$ and have also been used in field experiments (Ayres et al., 2015; Wagner et al., 2013; Sobanski et al., 2016).

In addition to the direct measurement of NO with the CL method, NO concentrations can be measured based on their absorption feature at 1,585.282 $cm^{-1}$ directly. For this method, a tunable

infrared laser differential absorption spectroscopy (TILDAS) instrument, utilizing an astigmatic multi-pass Herriott cell (Herndon et al., 2004) and a dual-wavelength spectrometer, based on a DFB laser emitting sequentially at 1,600 $cm^{-1}$ and 1,900 $cm^{-1}$ have been used for the measurement of the two species (Jagerska et al., 2015). The 1 s precision for NO measurement of the TILDAS instrument was 550 ppt, whereas that for the field experiments was 1.5 ppb. Thus, this technique may suffer from low

detection sensitivity compared with the CL method. Given the rapid changes of nighttime oxidation, i.e., $NO_3$ radical, understanding the rapid changes of its precursors, NO and $NO_2$ is thus a critical prerequired information to develop a nighttime atmospheric chemistry model. Conversion to $NO_2$ by adding excess $O_3$ can provide an indirect method for NO detection, which can achieve high sensitivity and high resolution (Fuchs et al., 2009; Wild et al., 2014).

The development of different technology provide the potential for NOx measurements on different platforms such as ground sites, vehicles as well as aircrafts (Yamamoto et al., 2011; Wagner et al., 2011; Castellanos et al., 2009). Due to the rapid economic growth in 2000-2010, China has become the second largest economy in the world. With the rapid growth of energy consumption, NOx emissions is increasing. Motor vehicles are one of the major sources for NOx, especially in urban areas

(Westerdahl, 2008). Exhaust from on-road vehicles makes up 20% of total NOx emissions in China (Shi et al., 2014). So a variety of methods have been used to measure the vehicle emissions to access air pollutant exposures and specifically impacts due to traffic-related emissions (Vogt et al., 2003; Carslaw and Beevers, 2004; Herndon et al., 2005; Lal et al., 2005; Burgard et al., 2006b; Hueglin et al., 2006; Burgard et al., 2006a; Wild et al., 2017). However, the methods are usually applied to monitor air

pollutants at several locations in large cities, the selection of which is critical for achieving representative measurements. The number of monitoring locations is not adequate to show the large scale patterns of the city. Hence, a direct on-road mobile instrument can be used to help obtain the spatial and temporal variations of NOx pollutants.



Here, we describe a dual-channel CRDS system based on the chemical conversion NO to NO$_2$ to
measure NO$_2$ and NO$_x$ simultaneously. In one channel, the sum of converted NO$_2$ from ambient NO
and ambient NO$_2$ is determined to provide a direct measurement of NO$_x$. In another channel, only
ambient NO$_2$ is measured. The subtraction of NO$_2$ measured in a second, independent channel provides
a direct measurement of NO alone. Measurements and comparison of NO$_2$ and NO between different
instruments were conducted to assess the accuracy of the dual-channel CRDS instrument. In addition,
the measurement of on-road vehicle emission plumes from the instrument during December 17, 2018 in
the region of Hefei, China was deployed. The main advantages of this instrument compared with CL
instruments are its low detection limit and high sensitivity as well as its potential ability for trace
measurements without calibration and interferences.

## 2. Setup of the instrument

Cavity ring-down spectroscopy has been applied to measurement NO$_3$ radical and N$_2$O$_5$ in our
group (Wang et al., 2015; Li et al., 2018; Li et al., 2018). In this work, the technique is applied to
measure NO$_2$ and NO. A schematic diagram of the dual-channel CRDS system developed in the present
work is shown in Fig. 1. The instrument mainly consists two identical CRDS systems for NO$_2$ and NO$_x$
detection, gas handling system, NO convertor and activated carbon device for NO$_x$ removing.

### 2.1. CRDS systems

A blue diode laser is used as the light source and the wavelength of the laser is monitored by a
spectrometer. The output of the diode laser with a center wavelength at 403.64 nm and a line width of
0.5 nm is directly modulated by a square wave signal (on/off) at a repetition of 2 kHz with a duty cycle
of 50% and the output power is about 60 mW. The light emitted by the laser first passes through an
isolator to prevent the reflected light into the laser and then enters into two identical cavities through
two reflecting mirrors and a 50/50 beam splitter. Each optical cavity is made of an aluminum tube with
an inner diameter of 9.4 cm. The two cavities are fixed rigidly by two frames, respectively. Two high
reflectivity mirrors are held in stable, adjustable mounts. The distance of two highly reflective mirrors
(LGR, 1 in. diameter, 1 m radius of curvature) is 75 cm for both channels. Consequently, ring-down
time constants in NO$_x$ and NO$_2$ cavities are 22.90 μs and 24.12 μs in dry N$_2$. The light emitted through
the back mirror of the cavity passes through a narrowband filter to filter stray light and then is directed
into a PMT. The signal passes through an amplifier and then enters into the digital acquisition card (NI
USB-6361, 16-bit, 2.0 Ms/s). The digital acquisition card is 1 MHz for each channel. Data are acquired
on the data acquisition board for a continuous period of 1.0 s during which 2000 decay traces are
transferred to the PC using a single transfer command and averaged to get a fitted decay trace at a laser
modulation rate of 2 KHz. The software algorithms calculate NO$_2$ concentration from, $\tau$ and $\tau_0$, the
ring-down time when the NO$_2$ is in the presence and absence of the cavity, respectively; the NO$_2$
absorption cross section, $\sigma$; the ratio of the total cavity length to the length over which the absorber is
present in the cavity, $R_L$ and the speed of the light, c. The concentration of the sample can be expressed
as follows:

$$[NO_2] = \frac{R_L}{c\,\sigma_{NO_2}}\ \left(\frac{1}{\tau} - \frac{1}{\tau_0}\right) \tag{1}$$

### 2.2. NO convertor

NO is measured by its conversion to NO$_2$ by adding excess O$_3$. The principle is based on the
following chemical equation (Sander et al., 2006).



$$NO + O_3 \rightarrow NO_2 + O_2 \quad k_1 \tag{R1}$$


Where $k_1 = 3.0 \times 10^{-12} \exp(-1310/T)$ cm$^3$ molec$^{-1}$ s$^{-1}$. Ozone is produced from $O_2$ photolysis at 185 nm by flowing 100 sccm of sampling air which is controlled by a MFC over a low pressure discharge mercury lamp. The mercury is inset into a quartz glass tube with a length of 50 mm and an inner diameter of 10mm. The flow rate passing through the mercury lamp was investigated and the resulting mixing ratio

of $O_3$ was detected by an $O_3$ analyzer (49i, Thermo Scientific). The $O_3$ concentration is approximately 11.2 ppm after mixing with the sampled air. A length of Teflon tubing (length 1 m, i.d. 3.8mm) serves as a reactor for the NO conversion.

### 2.3. Activated carbon device

Background measurement of $\tau_0$, that is the ring down time when the absorber is in the absence of

the cavity, is important for accurately retrieving the absorber concentration as well as for checking the cleanliness of the cavity mirrors. Usually zero air or chemical scrubber is used to acquire zeros (Wada and Orr-Ewing, 2005). In our system, zeros are obtained by passing sampled air through an activated carbon filled tubing with an outer diameter of 6.0 cm and a length of 26.0 cm through a three-way solenoid valve located below the filter holder. The $\tau_0$ is measured for 60 s every 10–16 min. This

frequency of zero measurements is observed to be sufficient to track drifts in the zero ring-down time constant measurement, with a stability of successive $\tau_0$ below 0.1% for 15 min intervals.

### 2.4. Gas handling system

The instrument gas handling system consists of sampling module and purge flow. The sampled air initially flows through a filter device loaded with filtering membrane (1 μm pore size) to prevent

light-scattering aerosols from entering the cavity with a rotary pump (K86KNE) and subsequently passes through the activated carbon device to provide the background measurement when the three-way solenoid valve is open or is directed toward the PFA tube when the three-way solenoid valve is closed. The air flow from the PFA tube is divided into three lines. Among which, 100 sccm sampled air, which is introduced into a quartz flow tube equipped with a mercury pen-ray lamp (Oriel 6035) to

generate $O_3$ by air photolysis as mentioned previously is merged with another 900 sccm sampled air and pulled into the $NO_x$ cavity. The third flow with a flow rate of 1 slm is directed into the $NO_2$ cavity. The flow rates of all of the gases are controlled by mass flow controllers. Each mirror is isolated from the sample flow by a purge volume that is continuously flushed with high-purity nitrogen at a rate of 25 ml min$^{-1}$ to prevent the degradation of the mirror reflectivity.

## 3. Results and discussion

### 3.1 Determination of Absorption Cross Sections

To retrieval the gas concentration, it is vital to determine the effective absorption cross section at peak absorption of the laser. The output waveforms of the laser, with a center wavelength of 403.64 nm and full width at half-maximum of 0.5 nm, was monitored by a spectrometer (QEPB0828) (red line

shown in Fig. 2).The center wavelength selected can cover the strong absorption of $NO_2$ and avoid the interference from other species, such as $H_2O$ (pink line in Fig. 2). The effective absorption cross section was determined to be $5.63 \times 10^{-19}$cm$^2$ /molecule by convolution the $NO_2$ absorption cross section by voigt (Voigt et al., 2002) with the laser spectrum (blue line in Fig. 2). A shift in the laser center wavelength would result in a change of the effective $NO_2$ cross-section. The day-to-day variability of



the laser center wavelength was less than 1% by monitoring the laser output for a few days. The largest uncertainty of the absorption cross section is about 3% according to voigt (Voigt et al., 2002).

### 3.2 The retrieval of $R_L$

Due to the purge gas to the mirrors, the $R_L$ value cannot be simply determined by the ratio of the distance between two mirrors to that between the inlet and outlet. The $R_L$ value was determined from

the absorption measurement of different concentrations of $NO_2$ ranging from 20 ppb to 70 ppb in the presence and absence of purge flow. The ratio of the two extinction measurements yielded a $R_L$ value independent of the $NO_2$ cross section and concentration. The $R_L$ value is determined to be $1.10 \pm 0.03$ for both the $NO_x$ and $NO_2$ channels.

### 3.3 The retrieval of $\tau_0$

In order to accurately determine the concentrations of trace gas by CRDS, it is very important to confirm background cavity loss measurements of $\tau_0$ when the target gases are not inside the cavity. Several alternative background measurement methods which incorporate zero air, a mixture of oxygen and nitrogen, chemically scrubbed laboratory air (using hydroxyapatite), and laboratory air sampled through the stainless steel tubing coil have been reported (Wada and Orr-Ewing, 2005) and each

experimental approach has its own merits and demerits. In our instrument, an activated carbon device was used for background measurement. The ring down times when the sampled air pass through the activated carbon device were determined to be $24.12 \pm 0.01$ μs and $22.90 \pm 0.01$ μs in two cavities, respectively. These values are close to those of measurements of zero air at the same sample rate for a 5 min period.

Two representative ring-down signals of the $NO_2$ from CRDS system when the $NO_2$ is in the presence and absence of the cavity are shown in Fig. 3. And the fitted ring down time were 24.12 μs and 20.30 μs respectively such that the $NO_2$ concentration is 20.28 ppb using the constants determined above.

### 3.4 Detection limit and measurement accuracy of two cavities.

The measurement precision of the dual-channel CRDS instrument for $NO_2$ and $NO_x$ detection was investigated with time series measurement of zero air (Fig. 4). The acquisition time for the spectral data was 1.0 s with an average of 2000 spectra. In order to analyze the stability of the instrument, the Allan variance had been calculated for the intensity measurements. For the two channels, the minima in the Allan plots indicated the optimum average times for optimum detection performance (right panel of Fig.

4) to be about 30 s. With 30 s integration time, the 1σ detection limits were 16 ppt and 14 ppt for the $NO_2$ and $NO_x$ channels, respectively.

The minimum detection can be written as follows:

$$[A]_{min} = \frac{\sqrt{2}R_L}{c\sigma}\left(\frac{\Delta\tau_0}{\tau_0^2}\right) \qquad (2)$$

For continuous zero $NO_2$ measurements, the $\Delta\tau_0$ was 0.008 μs in both $NO_x$ and $NO_2$ channels and

$\tau_0$ was 22.90 μs and 24.12 μs in $NO_x$ and $NO_2$ channels, respectively when averaging the data to 1 s. Taking the $R_L$ value to be 1.10 and $\sigma$ to be $5.63 \times 10^{-19}$ molecule/cm². The 1σ minimum detection limits determined from the previously presented equation for the $NO_x$ and $NO_2$ channels are 39 ppt and 35 ppt at an integration time of 1 s, respectively, which were close to the Allan variance analysis described above.

The total uncertainty of $NO_2$ measurement by CRDS was from the errors in $R_L$ and the $NO_2$





absorption cross section. The uncertainty in $R_L$ was less than 3%, and the uncertainty in the $NO_2$ absorption cross-section was about 4%. Considering all of these errors, the total uncertainty of $NO_2$ measurement was determined to be 5%.

The minimum detection and uncertainty of our instrument is further compared with the existing field measurement techniques for $NO_2$ measurements. (Table 1).

### 3.5 NO Conversion Efficiency

The main factor determining the NO conversion efficiency was the flow rate passing through the mercury pen-ray lamp which therefore influences the generated $O_3$ concentration. The mixing ratio of $O_3$ in the $NO_x$ channel line changing with the flow rate that passes through the mercury pen-ray lamp was investigated and the result was shown in Fig. 5. As a result, the bypass flow passing through the Hg lamp was determined to be 100 sccm. Under this condition, when the residence time of $O_3$ in the cavity is 1s and ambient $NO_2$ concentration is 50 ppb, NO conversion efficiency with different NO concentrations (10-1000 ppb) is simulated and NO conversion efficiency is larger than 98% .

Because the cross section of $O_3$ is about four orders magnitude of smaller than that of $NO_2$ at the center wavelength of the laser, the absorption of $O_3$ generated by mercury photolysis is negligible. According to Fuchs (Fuchs et al., 2009), under conditions when NO abundance is rich, further oxidation of $NO_2$ to $NO_3$ and $N_2O_5$ has only a slight effect on $NO_x$ measurement, such that correction of the $NO_x$ measurement can be neglected. However, under conditions when NO is absent, the loss of $NO_2$ due to oxidation by high concentration of ozone is indeed one of the main factors that attributes to the errors in the $NO_x$ channel. The reaction equation is expressed as follows:

$$NO_2+O_3 \rightarrow NO_3+O_2 \qquad k_2 \qquad \text{(R2)}$$

$$NO_3+NO_2 \leftrightarrow N_2O_5 \qquad keq \qquad \text{(R3)}$$

where $k_2 = 1.2 \times 10^{-13} \exp(-2450/T)$ cm$^3$ molec$^{-1}$ s$^{-1}$ (T=298K, $k_2 = 3.2 \times 10^{-17}$ cm$^3$ molec$^{-1}$ s$^{-1}$)(Sander et al., 2006), $keq = (5.1 \pm 0.8) \times 10^{-27} \exp(10871/T)$ cm$^3$ molec$^{-1}$ s$^{-1}$ (T=298K, $k_{eq} = 3.5 \times 10^{-11}$ cm$^3$ molec$^{-1}$ s$^{-1}$) (Osthoff et al., 2007) respectively. The loss rate will increase with the increase in the $NO_2 + O_3$ reaction rate constant when temperature in the cavity increases. Moreover, the loss rate is sensitive to the $NO_2$ mixing ratio. Diluted $NO_2$ standard mixture was introduced into two channels to characterize the effect of high concentration ozone on $NO_2$ measurement. The $NO_2$ concentrations and the correlation plot between data in the two channels are shown in Fig. 6. The interference of $O_3$ in $NO_x$ channel when NO is absent can be neglected. The discrepancy between two different channels may be caused by the systematic errors in two different channels and can be corrected with the coefficient obtained from Fig. 6 (b).

## 4. Field applications

### 4.1 Standard mixtures of NO and NO₂ measurement.

The comparisons of $NO_2$ measurements between CRDS and $NO_x$ analyzer have been carried out on $NO_2$ standard mixtures. Different mixing ratios of $NO_2$ were obtained by gas phase titration of NO with excess $O_3$, which was generated by an ozone generator (OC500). The 10.3 ppm NO standard mixture was initially diluted by $N_2$ and subsequently oxidized by $O_3$. The amount of $NO_2$ generated





from excess ozone can be calculated from the known initial concentration of NO. The generated pure
290    $NO_2$ standards in clean air were in the concentration range of 20-70 ppb. The CL analyzer used for
comparison in this laboratory experiment was separately calibrated and the linearity of this instrument
was checked using a mixture containing NO. Fig. 7 (a) shows the concentration of standard $NO_2$ in the
laboratory measured by CRDS and simultaneously by a commercial CL analyzer (42i, Thermo
Scientific, Inc., 0.4 ppb (1σ) detection limit). A correlation analysis between data from the two
295    instruments was carried out. The fitting results shown in Fig. 7(b) indicate that $NO_{2\,(CRDS)}$ = $NO_{2\,(CL}$
$_{analyzer)}$ × 1.031 - 0.940, with a linear correlation factor ($R^2$) of 0.99. The results in Fig. 7 (a) also
indicates that CRDS instrument can capture the $NO_2$ variation more rapidly than CL analyzer.

### 4.2 Ground-based measurements of $NO_2$ and NO.

The $NO_2$ concentration measured by the dual-channel CRDS instrument was compared with the
300    results obtained by a CEAS instrument (Duan et al., 2018) during the period from November 3 to 5,
2017 in the western suburb area of Hefei, Anhui, China. The CEAS instrument not the CL analyzer was
selected for $NO_2$ intercomparison because the CL analyzer must convert $NO_2$ to NO, exposing itself to
chemical interferences whereas the CEAS instrument directly detects $NO_2$. Measurement precisions
(1σ) for $NO_2$ is about 170 ppt in 30 s. The time resolution of CRDS and CEAS instruments are 1s and
1min respectively. The CRDS and the CEAS instruments were setup on the sixth floor of the building
but apart by tens of meters in Anhui Institute optic and Fine mechanics. The area directly (1–1.5 km) to
the northeast and the south of the site is the Dongpu reservoir. The area in the northwest to north sector
is surrounded by a mix of trees. The significant $NO_2$ pollution directly found during the measurement is
the emission of the cars along the road (100 m radius). The air originated from the sector between the
South and East (5 km) may bring the anthropogenic emission to the site. Ambient air was introduced
into the instruments by use of a 6 mm outer diameter Teflon tube. The inlet of the Teflon tube was
outside the building through the window. The data for comparison were averaged to 1min. Fig. 8 (a)
shows the temporal variations of $NO_2$ concentrations measured by the CEAS and CRDS instruments.
The nighttime $NO_2$ was in the range of 35 ppb to 3 ppb. The $NO_2$ concentrations and variations
measured by the CRDS instrument were consistent with those measured by the CEAS instrument. The
least-squares analysis showed that the slope and intercept of the regression line were 1.042 ±0.002 and
(−0.393 ± 0.040) ppb, respectively as shown in Fig. 8. However, the results revealed a discrepancy
where rapid $NO_2$ variations appeared. We attribute this discrepancy to the slight difference between the
two inlets of the instruments when large $NO_2$ was rapidly injected into the atmosphere. In general, the
CRDS instrument has substantive advantages for retrieving rapid variations of $NO_2$ plums due to its
high time resolution and high sensitivity.

The comparison of NO concentrations measured by the dual-channel CRDS instrument and CL
analyzer was conducted under a variety of sampling conditions for a total of seven days at the site
described previously. Both instruments were attached to the same air sample inlet. The data sets from
the CRDS instrument and CL analyzer were highly correlated over wide concentration ranges of NO.
Fig. 9 (b) shows the relationship between NO concentrations observed by the CRDS and CL methods.
The slope and intercept of the regression line were 0.959 ± 0.007 and 0.352 ± 0.013 ppb. The
correlation coefficient is $R^2$=0.99. The CL analyzer is capable of measuring NO reliably. Therefore, the
dual-channel CRDS instrument is also considered to be a reliable method for the measurement NO.

### 4.3 On-road measurements of vehicle $NO_2$/NOx emission.

In order to retrieval the vehicle emissions on road, field measurements were performed in Hefei



from 15:00 to 16:00 CST on 17 December 2018. The CRDS instrument was powered by a lithium
battery, and ambient air was pumped into the system though an inlet fixed on the roof of the car. The
vehicle speed is about 50 km/h. In order to get the discrepancy of vehicle emissions in urban and
suburban areas, the car travels along these areas. Fig. 10 shows a picture of the movable van loaded
with CRDS instrument and the position of the sampling inlet, about 1.5 m above ground.   illustrates
the route in Hefei and the drive track is colored logarithmically with respect to measured $NO_x$、$NO_2$ and
NO. The $NO_2$ concentration ranged from 1.5ppb to 133.3 ppb and NO ranged from detection limit to
554.7 ppb respectively. The mean concentrations of NO and $NO_2$ were 140 ppb and 54.9 ppb,
respectively. NO and $NO_2$ concentration were higher in urban area than in suburban area. Large plumes
of NO were found at the crossroads with heavy traffic or converged with heavy-duty diesel vehicles.
$[NO_2]/[NOx]$ ratio was about 19%, a number that is larger than the results in USA (Wild et al., 2017).
Because The $NO_2$ to NOx emissions ratio affects ozone production and spatial distribution, more
efforts should be done to provide a constraint on emissions inventories used in air quality modeling.
The mobile CRDS instrument provides a good method to retrieval the direct vehicle NOx emission and
plume $NO_2$ to NOx ratio due to its easy deployment and high temporal resolution.

## 5. Conclusion

Demonstration of a compact, sensitive, and accurate instrument for detection trace amounts of
$NO_2$ and $NO_x$ in ambient air has been achieved by using diode-laser cavity ring-down spectroscopy
with the center wavelength of 403.64 nm. Minimum detection limits of $NO_2$ and $NO_x$ were estimated to
be 0.030 ppb and 0.040 ppb at an integration time of 1s when zero air is sampled with measurement
accuracy of ±5%. Measurements of $NO_2$ using dual-channel CRDS instrument and CL analyzer on
standard mixtures were performed in the present work which demonstrated a good correlation between
these techniques. In order to confirm the reliability of the dual-channel CRDS instrument in the field
atmosphere. Continuous measurement was conducted and the stability of the instrument was shown.
When comparing the dual-channel CRDS instrument for $NO_2$ measurement with a CEAS instrument
and NO measurement with the CL analyzer in the field, the results both showed a good correlation.

The CRDS instrument was deployed in a movable car to monitor NO and $NO_2$ emission on road.
The advantage of high time resolution of the instrument can provide a direct method for on-road
vehicle plumes measurement. Meanwhile, the instrument has high detection sensitivity, which can also
provide a new detection technique for chemistry model verification. The instruments developed could
lead to the wider application for ambient air quality monitoring and will be useful to investigate
photochemistry in the atmosphere more precisely.

## Acknowledgments

This work was supported by National Natural Science Foundation of China (61575206, 91644107,
41571130023and 61805257) and the National Key Research and Development Program of China
(2017YFC0209401, 2017YFC0209403).

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



**Table 1    Comparison of NO$_2$ detection limits based on optical methods.**

| Principle of measurement | Laser power | Wavelength range/nm | Dectection limit | Reference |
|---|---|---|---|---|
| Cw-CRDS | 5mW(1MHz) | 410 | 80ppt/50s | (Wada and Orr-Ewing, 2005) |
| ND:YAG laser CRDS | 1mJ | 532 | 40ppt/1s | (Osthoff et al., 2006) |
| pDL-CRDS | 40 mW(2KHz-10%) | 404 | 22ppt/1s(2σ) | (Fuchs et al., 2009) |
| Fabry-Perot (FP) pDL-CRDS | 1.1w (4KHZ-10%) | 400 | 38ppt/128ms | (Karpf et al., 2016) |
| commercial DL-CRDS | 1.2KHZ | 407.38 | 60ppt/60s(3σ) | (Castellanos et al., 2009) |
| LED-based commercial CRD | 355 mW | 397-412(405) | 80ppt/60s | (Brent et al., 2013) |
| LED-CEAS | 340mw | 455 | 2.2ppb/100s(1σ) | (Wu et al., 2009) |
| Xe lamp DOAS | | 295–492 nm | 2ppb/8-12min | (R. McLaren, 2010) |
| CAPS | | 440 | 60ppt/10s(3σ) | (Kebabian et al., 2008) |
| diode-pumped Nd:YAG laser-LIF | 15mw(14KHz) | 473 | 140/60s | (Taketani et al., 2007) |
| blue LED-IF | 17.7mw(10KHz) | 435 | 9.8ppb/60 | (Matsumi et al., 2010) |
| pulsed blue light LED-LIF | 22mw | 430 | 7ppb/1min | (Sadanaga et al., 2014) |
| pDL-CRDS | 60mw | 403.64 | 30ppt/1s | This work |

CRDS=cavity ring-down spectroscopy; CEAS=cavity-enhanced absorption spectroscopy; BB=broadband; DOAS=differential

optical absorption spectroscopy; cw=continuous-wave diode laser. LIF=laser induced fluorescence; CAPS= cavity attenuated

phase shift spectroscopy; pDL=pulsed diode laser.





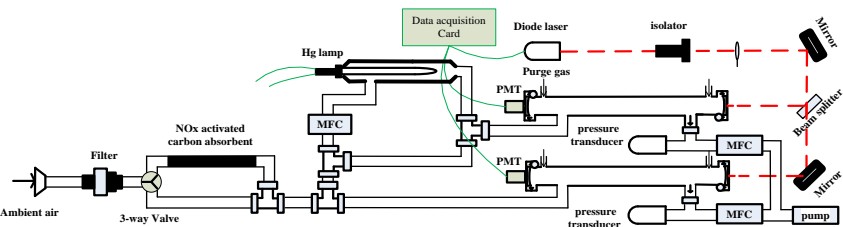

585                                    Fig. 1. Schematic of dual-channel Cavity Ring down Spectroscopy system.

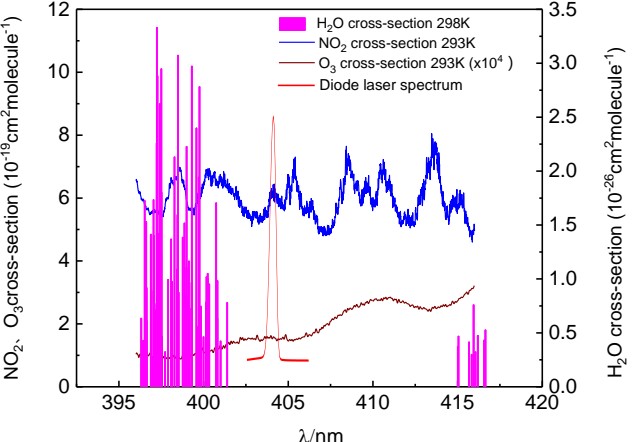

Fig. 2. Cross section of the NO₃ radical, NO₂, O₃, water vapor, and diode laser spectrum.

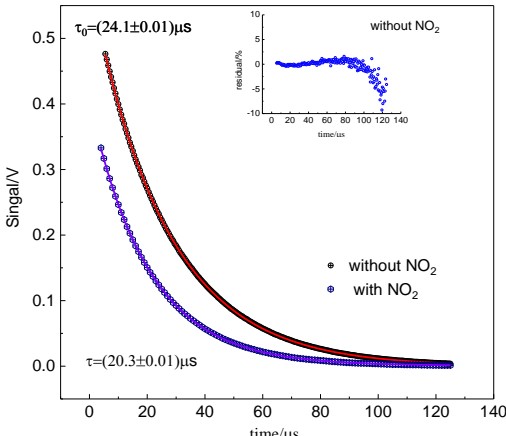

590                   Fig. 3. Different cavity ring-down signals and fitting results in the absence and presence of NO₂. The

small figure in the upper right corner is the fitting residual.





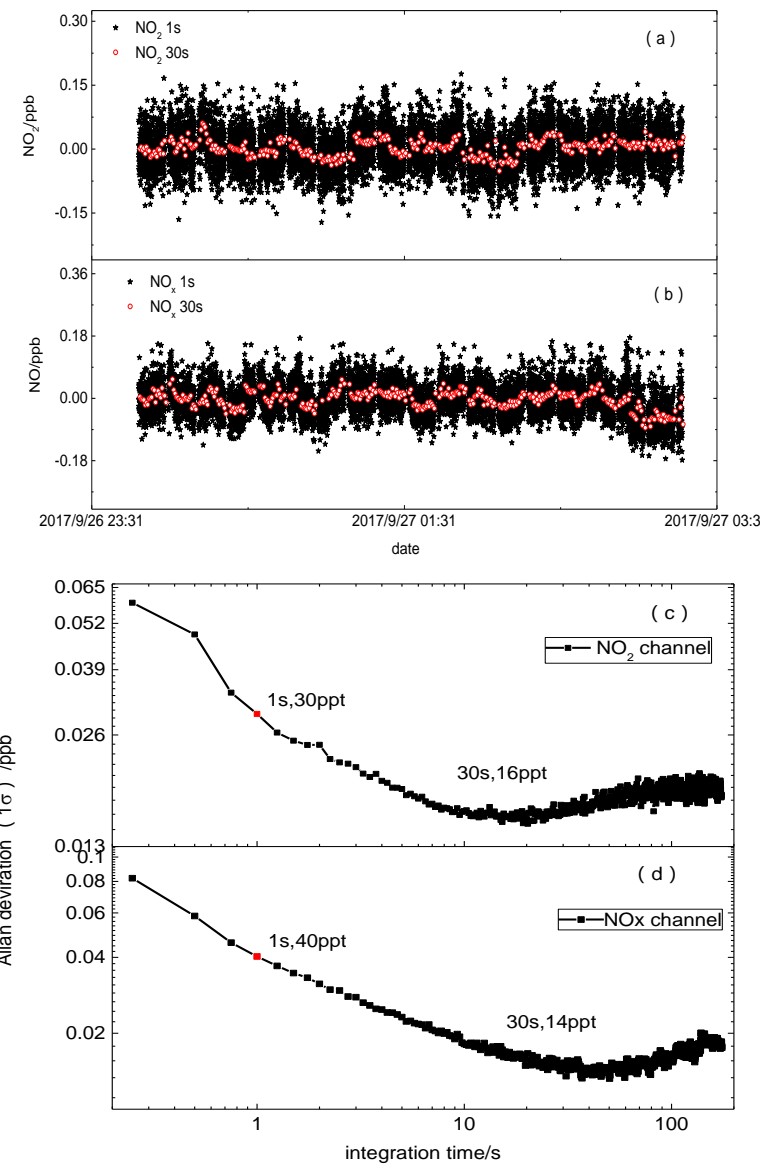

Fig. 4. (a)(b) Continuous time series measurement when the instrument sampled only zero air, averaged to 1s for

NO₂ and NOₓ channels (black dots), the red dots show the data averaged to 30s; (c)(d): Allan deviation plots for

NO₂ concentration in two channels. The minimum value equals the optimum integration time.



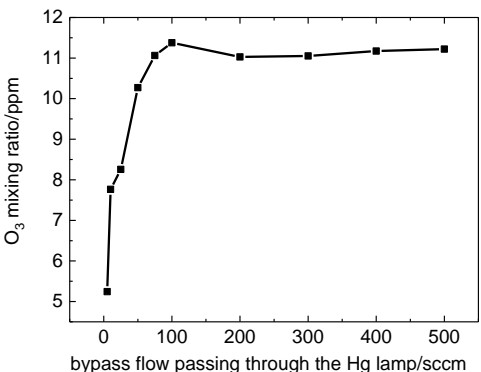

Fig. 5. O$_3$ mixing ratio when changing the bypass flow passing through Hg lamp.


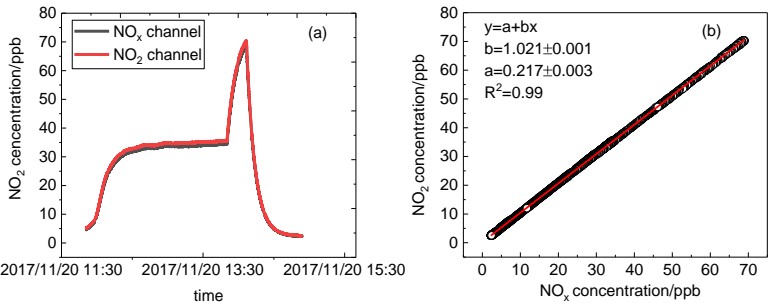

Fig. 6. (a) Time series of NO$_2$ concentration sampled standard mixtures by CRDS instrument in two channels with mercury pen-ray lamp switched on. (b) A correlation plot between the data from two channels.


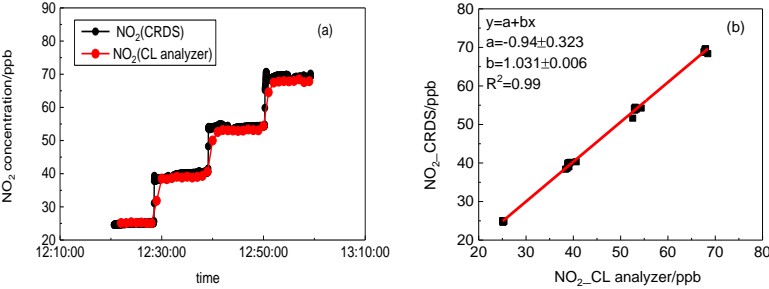

Fig. 7. (a) Time series of NO$_2$ concentration sampled standard mixtures by CRDS instrument and CL analyzer. The time resolution for CRDS instrument and CL analyzer are 1s and 1min, respectively. (b) A correlation plot between the data from the CRDS instrument and the CL analyzer (data averaged to 1min). The fitting result gave a gradient of 1.031 and an intercept of -0.940 ppb, with linear correlation factor of 0.99.






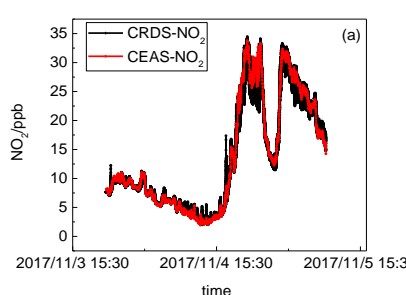
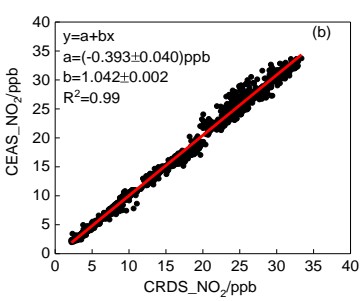

Fig. 8. (a) NO$_2$ mixing ratios by CEAS (1 min average) and CRDS (1 s average) instruments, (b) Scatter plots for the NO$_2$ dataset

615     from CRDS and CEAS instrument. The red lines illustrate the linear regression (Data averaged to 1 min base).

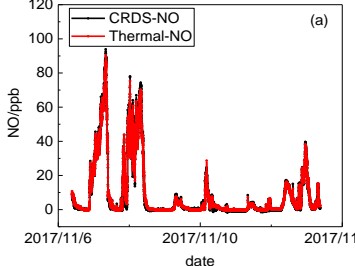
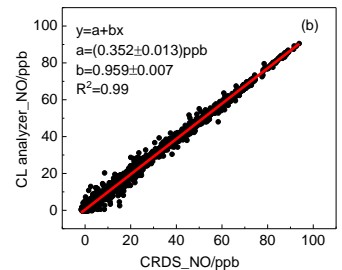

Fig. 9. (a) Time series of NO by dual-CRDS instrument and CL analyzer. (b)A correlation between two

instruments is shown and data for correlation analysis is averaged in 1min.


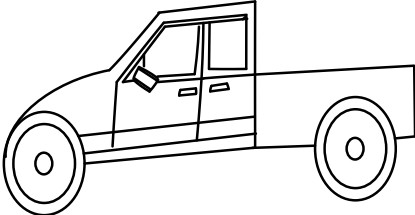
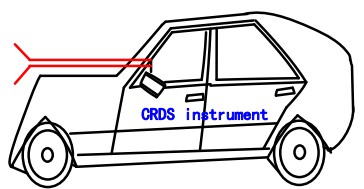

Fig. 10 The diagram of the movable van loaded with CRDS instrument.




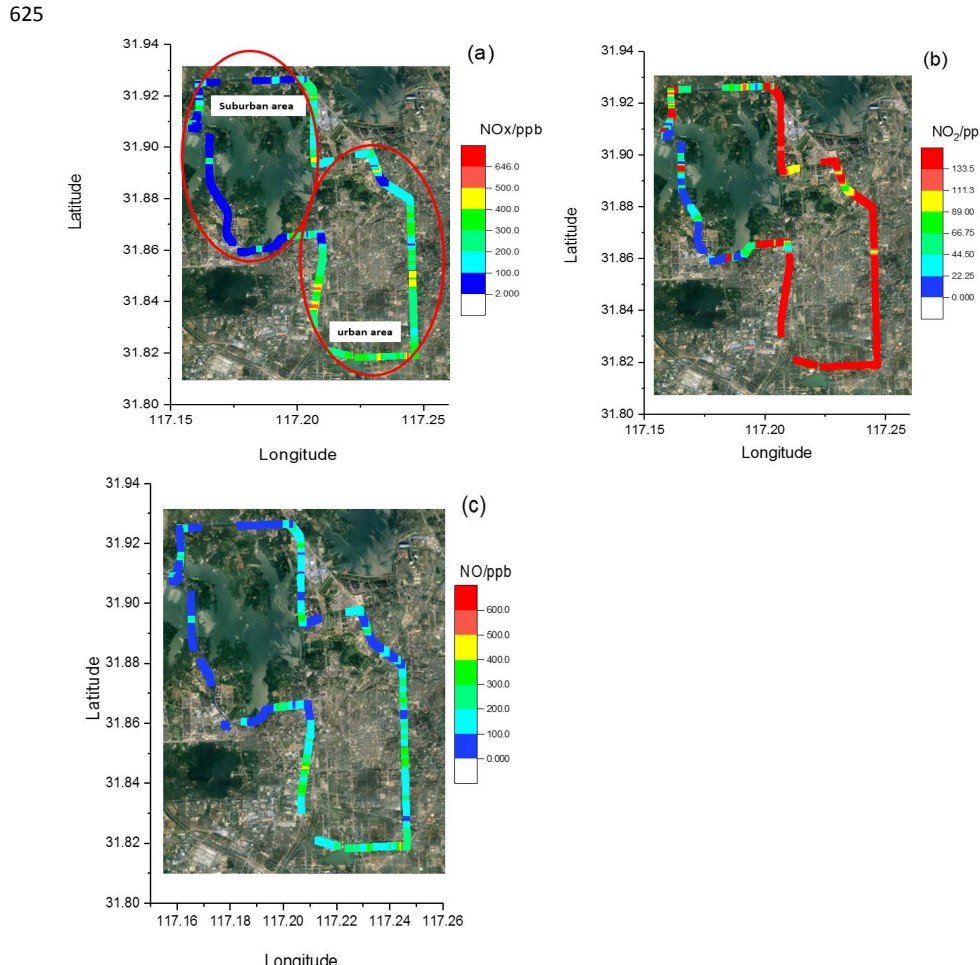

Fig. 11. Results of the NOx (a)、 NO₂ (b)、 NO (c) concentrations around Hefei, China (Data is averaged to 5s).