# Peer review of "Simultaneous measurement of NO and NO2 by dual-channel cavity ring down spectroscopy technique"

_Atmospheric Measurement Techniques, 2018_

## Referee Comment (RC1) · Hongbing Chen (Referee) · 23 Feb 2019

Could authors provide more test data on-road measurements of vehicle NO2/NOx emissions? And if possible, the on-road intercomparsions will be meaningful for the validation of the method and instrument.

It will be helpful if someone can review and make some corrections the English writing.

page 1, line 23: " Too much NOx are ......" too much is how much? "are" to "is" page 2, line 55 "too much high"

page 15, table 1: it will be better to convert the detection limits of all different measure-

ments to an unified unit such as ∼"ppt/1s(1sigma)"

---

## Referee Comment (RC2) · Anonymous Referee #2 · 12 Mar 2019

The paper describes a new instrument to measure NO and NO2 with high sensitivity and high time resolution for future applications on board of a vehicle. The technique is based on CRDS using a laser diode at 403 nm and contains two pathways: one measuring NO2 and the other measuring NOx after oxidation of NO to NO2 by addition of O3. The technique is not new and has already been applied before to NO and NO2 measurements with similar sensitivities and time resolution (Fuchs et al. 2009). The reliability of the instrument has been demonstrated by comparing NOx and NO2 measurements for several days with CL and CEAS instruments. A first demonstration of real-time measurements onboard a vehicle is shown for a 1 hour ride through Hefei. While the paper does not give a brand new idea, it is a solid description of a new

instrument that can be useful in future applications. The manuscript could be accepted to AMT after some English polishing. I have a few minor points:

I don't think NOx is not a secondary pollutant. Unfortunately, I have no access to the paper you cite (Crutzen 1979) in order to confirm, but I invite you to re-verify this paper to see on what basis Crutzen said that NOx is a secondary pollutant. Also, from NOx to the formation of secondary aerosols it is a long stretch, maybe it's better to remove.

Page 2, line 78: indicating several seconds in combination with high sensitivity is strange: do you mean a low detection limit with several seconds time resolution?

You give several times your ring-down times with 2-digit precision (24.12 and $22.90\mu$s), however mirrors get polluted, alignment changes etc, so I guess the ring-down times you have given here are the result of one measurement at one moment? Or do you really measure over several days or weeks always exactly the same ring-down time?
* * *

---

## Author Comment (AC1) · 26 Mar 2019

Thanks for the reviewer's questions. The manuscript will be revised following the reviewer's suggestions.

1. Could authors provide more test data on-road measurements of vehicle NO2/NOx emissions? And if possible, the on-road intercomparsions will be meaningful for the validation of the method and instrument.

Reply: The intercomparsion between the CRDS instrument with NO analyzer for NO measurements and the intercomparsion between the CRDS instrument with CEAS

instrument for NO2 measurements have been done to valid the accuracy of the instrument in the paper, so we don't provide other data on road measurements of vehicle NO2/NOx emissions.Based on the reviewer's suggestions, we performed another measurement of vehicle emissions on road. NO analyzer (42i), O3 analyzer (49i) and the CRDS instruments were placed in a car and were powered by three batteries. Ambient air was pumped through an inlet fixed on the roof of the car and then was divided into three lines to the instruments, respectively. Fig. 1 illustrates the 4-hour drive around Hefei, the drive track involve highway, urban and suburban area and is colored with respect to the measured NO and NO2. Vehicle speeds varied greatly on the three different road types and vehicle speed is around 100km/h on highway. Influenced by the vehicle emissions, the NOx plumes on urban roads are higher than those on suburban roads and highway. Fig.2 shows the time series of NO2,NO and O3.O3 and NO showed a significant negative correlation and O3 can be titrated by NO quickly. Fig.3 shows the NO data measured by CRDS and CL analyzer (42i), (a) is the raw data for the CRDS instrument and (b) is the data with time resolution of is 1 min for the CRDS instrument. The good agreement between the two instruments proves that the CRDS instrument can be applied for fast vehicle NOx emissions.

It will be helpful if someone can review and make some corrections the English writing. page 1, line 23: " Too much NOx are ......" too much is how much? "are" to "is" page 2,line 55 "too much high"

Reply: Since the questions above are about the English expression and grammar mistakes, the revisions will be done in the final version.

page 15, table 1: it will be better to convert the detection limits of all different measurements to an unified unit such as "ppt/1s(1sigma)

Reply: The suggestion will be followed in the revised manuscript.

[Figure]

Fig.1.The 4-h drive around Hefei, China, colored by the measured NO (a) and NO$_2$ (b)

concentrations, respectively.

**Fig. 1.**

[Figure]

Fig.2. Results of the NO$_2$、NO and O$_3$ concentrations around Hefei, China.

**Fig. 2.**

[Figure]

Fig.3. Time series of NO by CRDS instrument and CL analyzer (42i). (a) is the raw data for the

CRDS instrument; (b) is the data with time resolution of is 1 min for the CRDS instrument.

**Fig. 3.**

---

## Author Response (AR1)

**Response to the Reviewers' comments on the manuscript:**

**Simultaneous measurement of NO and NO₂ by dual-channel cavity ring down spectroscopy technique**

rzhu@aiofm.ac.cn

Thanks for the reviewer's questions. The manuscript will be revised following the reviewer's suggestions.

**Reviewer # 1**

**Comments and suggestions:** 1. Could authors provide more test data on-road measurements of vehicle NO₂/NOx emissions? And if possible, the on-road intercomparsions will be meaningful for the validation of the method and instrument.

**Reply:** The intercomparsion between the CRDS instrument with NO analyzer for NO measurements and the intercomparsion between the CRDS instrument with CEAS instrument for NO₂ measurements have been done to valid the accuracy of the instrument in the paper, so we don't provide other data on road measurements of vehicle NO₂/NOx emissions. Based on the reviewer's suggestions, we performed another measurement of vehicle emissions on road. NO analyzer (42i), O₃ analyzer (49i) and the CRDS instruments were placed in a car and were powered by three batteries. Ambient air was pumped through an inlet fixed on the roof of the car and then was divided into three lines to the instruments, respectively. Fig. 1 illustrates the 4-hour drive around Hefei, the drive track involve highway, urban and suburban area and is colored with respect to the measured NO and NO₂. Vehicle speeds varied greatly on the three different road types and vehicle speed is around 100km/h on highway. Influenced by the vehicle emissions, the NOx plumes on urban roads are higher than those on suburban roads and highway.

Fig.2 shows the time series of NO₂, NO and O₃.O₃ and NO showed a significant negative correlation and O₃ can be titrated by NO quickly.

Fig.3 shows the NO data measured by CRDS and CL analyzer (42i), (a) is the raw data for the CRDS instrument and (b) is the data with time resolution of is 1 min for the CRDS instrument. The good agreement between the two instruments proves that the CRDS instrument can be applied for fast vehicle NOx emissions.

[Figure]

[Figure]

Fig.1.The 4-h drive around Hefei, China, colored by the measured NO (a) and NO$_2$ (b) concentrations, respectively.

[Figure]

Fig.2. Results of the $NO_2$、 NO and $O_3$ concentrations around Hefei, China.

[Figure]

Fig.3. Time series of NO by CRDS instrument and CL analyzer (42i). (a) is the raw data for the CRDS instrument ; (b) is the data with time resolution of is 1 min for the CRDS instrument.

**Comments and suggestions:** It will be helpful if someone can review and make some corrections the English writing.
Page 1, line 23:" Too much NOx are ......" too much is how much? "are" to "is" page 2, line 55 "too much high"
**Reply:** Since the questions above are about the English expression and grammar mistakes, the revisions will be done in the final version.

**Comments and suggestions:** page 15, table 1: it will be better to convert the detection limits of all different measurements to an unified unit such as ~"ppt/1s (1sigma)
**Reply:** The suggestion will be followed in the revised manuscript.

**Reviewer # 2**

**Comments and suggestions:** I don't think NOx is not a secondary pollutant. Unfortunately, I have no access to the paper you cite (Crutzen 1979) in order to confirm, but I invite you to re-verify this paper to see on what basis Crutzen said that NOx is a secondary pollutant. Also, from NOx to the formation of secondary aerosols it is a long stretch, maybe it's better to remove.

**Reply**: In order to avoid the ambiguity caused by this sentence, I will rewrite it although I think that $NO_2$ is a secondary pollutant. And I agree with the reviewer's idea that from NOx to the formation of secondary aerosols it is a long stretch, I will remove it.

**Comments and suggestions:** Page 2, line 78: indicating several seconds in combination with high sensitivity is strange: do you mean a low detection limit with several seconds time resolution?

**Reply**: Yes, I mean that the optical methods can achieve a high detection sensitivity and the detection limit is several ppt level with several seconds time resolution.

**Comments and suggestions:** You give several times your ring-down times with 2-digit precision (24.12 and 22.90 μs), however mirrors get polluted, alignment changes etc, so I guess the ring-down times you have given here are the result of one measurement at one moment? Or do you really measure over several days or weeks always exactly the same ring-down time?

**Reply**: The ring-down times I have given here are the results of one measurement and the ring-down times are used to calculate the detection limit in both channels. However, in field measurements, the ring-down times when the targeted gases are not in the cavity ($\tau_0$) are measured every 10 min for a time period of 1min, the measurement results show that the ring-down times during the 10-min interval change slightly. The ring-down time will change after measuring over several days or weeks influenced by temperature, pressure, alignment changes etc. This may slightly affect the detection limit of the instrument but can not affect the concentration retrieval.

**The list of all relevant changes made in the manuscript**

1, P1. Line 22-23.

2, P2. Line 51-53 and Line 75-77.

3, P4. Line 151-152.

4, P6. Line 223-225.

5, P9. Line 339-354.

6, P15. Table 1.

7, P21-22. Line 635-641.

[revised manuscript text omitted]